# A Hidden Chaotic System with Multiple Attractors

**DOI:** 10.3390/e23101341

**Published:** 2021-10-14

**Authors:** Xiefu Zhang, Zean Tian, Jian Li, Xianming Wu, Zhongwei Cui

**Affiliations:** 1Institute of Advanced Optoelectronic Materials, Technology of School of Big Data and Information Engineering, Guizhou University, Guiyang 550025, China; zhangxiefu@gznc.edu.cn; 2College of Mathematics and Big Data, Guizhou Education University, Guiyang 550018, China; lijian@gznc.edu.cn (J.L.); zhongweicui@gznc.edu.cn (Z.C.); 3College of Computer Science and Electronic Engineering, Hunan University, Changsha 410082, China; 4School of Mechanical and Electrical Engineering, Guizhou Normal University, Guiyang 550025, China; jsdxwxm@126.com

**Keywords:** multiple attractors, hidden attractors, Spectral Entropy, coexistence attractors, transition behavior, electronic circuit

## Abstract

This paper reports a hidden chaotic system without equilibrium point. The proposed system is studied by the software of MATLAB R2018 through several numerical methods, including Largest Lyapunov exponent, bifurcation diagram, phase diagram, Poincaré map, time-domain waveform, attractive basin and Spectral Entropy. Seven types of attractors are found through altering the system parameters and some interesting characteristics such as coexistence attractors, controllability of chaotic attractor, hyperchaotic behavior and transition behavior are observed. Particularly, the Spectral Entropy algorithm is used to analyze the system and based on the normalized values of Spectral Entropy, the state of the studied system can be identified. Furthermore, the system has been implemented physically to verify the realizability.

## 1. Introduction

Since the first chaotic system with a hidden attractor in Chua’s system was discovered [1], hidden attractors have motivated great interest because of their importance in both theory and engineering. In the chaotic system, there are two kinds of attractors classified by Kuznetsov et al. [2]: self-excited attractor [3] and hidden attractor. If the basin of attraction is associated with an unstable equilibrium, it is a self-excited attractor, if the basin of attraction does not intersect with small neighbourhoods of any equilibrium, it is the hidden attractor [4,5]. There are various types of hidden chaotic attractors such as the attractors with stable equilibrium points [6,7,8], with lines or curves of equilibrium points [9,10], with surfaces of equilibrium points [11] or without equilibrium points [12,13,14].

Multi-stability is an important phenomenon, meaning an infinite number of attractors generated through varying the initial values or system parameters. Pham et al. investigated the multi-stability of a novel hidden chaotic system without equilibrium [13] Yang et al. proposed a possible approach to construct a hidden system with infinitely many stable equilibria and infinitely many chaotic attractors [15]. Rajagopal et al. constructed a new cyclic symmetry chaotic system which presents multi-stability in an interval of its system parameter [16]. Varshney et al. found a large number of periodic hidden attractors and analysed the reasons of multi-stability [17]. Generally, the chaotic system with the symmetry is vulnerable to multi-stability [13,16,17,18,19]. Moreover, Transition behavior is also an important research topic and has recently received much attention. Ma et al. observed transition behavior from chaos to weak chaos [20] and Liu et al. found the transient behavior from weak chaos to quasi-period [21]. Du et al. proposed a hidden system with multiple transient transition behaviors and two to five different states have been observed during the transient process [22]. Due to their dynamic characteristics, the chaotic systems have found their applications in various fields, such as physics [23,24], bio-robotics [25,26], secure communication based on chaos synchronization [27,28], economics [29,30], etc.

Inspired by [13], a hidden chaotic system with multiple attractors is constructed and some attractive dynamic characteristics are observed. First, seven types of attractors are obtained, such as chaotic attractor, quasi-periodic attractor, periodic attractor and hyperchaotic attractor through altering system parameters. Moreover, when changing some parameters, the attractors are controllable, both in location and shape. Through changing the initial values of the system, the phenomenon of the coexistence attractor is presented. Furthermore, under certain conditions, transition behavior can be discovered. By comparison with Viet-Thanh Pham and Christos Volos’s system [13], the proposed system can generate seven types of attractors such as periodic, quasiperiodic and chaotic attractors. Besides, the hyperchaotic behavior and transition behavior have been observed, demonstrating better complexity.

In this work, the studied system is solved by the fourth-fifth order Runge-Kutta method and the dynamic characteristics of the hidden system are analyzed by several numerical methods, including phase diagrams, Largest Lyapunov exponent (LLE), bifurcation diagram, time domain diagram, attractive basin and Spectral Entropy (SE). The SE algorithm especially is used to show the characteristics of the hidden system. Recently, the SE algorithm has been used to analyze the dynamic characteristics of chaotic systems. Zolfaghari-Nejad et al. confirmed the period-subtracting phenomenon through SE distribution diagram [31]. Wang et al. analyzed the basic dynamics of a fractional-order chaotic system with a hidden system by comparing LLE and SE algorithms in [32], and the SE algorithm was also used to exhibit multi-stability under different initial values [33,34,35]. In this paper, we give the specific normalized average values, based on which the state of the studied chaotic system can be identified. Recently, the significance of the implementation with real case electronic circuits has been studied [36]. The proposed system is realized by the circuit and the experimental results are consistent with the numerical simulation results that testify the feasibility.

The rest of this paper is arranged as follows. Section 2 gives a description of the mathematical model. In Section 3, we investigate the nonlinear dynamic characteristics of the studied system. Section 4 verified the theoretical analysis through the circuit experiments. Section 5 summarizes the whole work and Section 6 discusses the limitations of the SE and LLE algorithm.

## 2. System Descriptions

### 2.1. The Description of the New System’s Mathematical Model

A three-dimension chaotic system proposed by [13] has been described as:(1)$$\left\{\begin{array}{l} \overset{•}{x}=y\\ \overset{•}{y}=-x-yz\\ \overset{•}{z}=\left|x\right|+xy-a \end{array}\right.,$$
where *x*, *y* and *z* are state variables and *a* is non-zero parameter. By adding a variable *w* in the above system, a novel four-dimension chaotic system is proposed as described below.
(2)$$\left\{\begin{array}{l} \overset{•}{x}=ky\\ \overset{•}{y}=-x-yz\\ \overset{•}{z}=\left|x\right|+xy-a\\ \overset{•}{\omega }=x-bw \end{array}\right.,$$

Here *x*, *y*, *z*, and *w* are state variables and *a*, *b*, *k* are non-zero parameters. The system is symmetrical with the transformation (*x*, *y*, *z*, *w*) → (−*x*, −*y*, *z*, −*w*). To get the equilibrium points of the proposed system, we need to solve the roots of Equation (3).
(3)$$\left\{\begin{array}{l} ky=0\\ -x-yz=0\\ \left|x\right|+xy-a=0\\ x-bw=0 \end{array}\right..$$

However, Equation (3) obviously has no solution, thus there is no equilibrium point. By definition, there are only hidden attractors whose basin of attraction does not intersect with small neighborhoods of any unstable equilibrium [6,7].

### 2.2. Description of Chaotic Behavior

In this subsection, some numerical methods are used to discuss the chaotic behavior of the proposed system, including phase diagrams, Lyapunov exponent spectrum (*LEs*) and the Poincaré map. All simulations are carried out with a constant parameter of *a* = 1.35, *b* = 1, *k* = 1 and the initial condition of (*x*(0), *y*(0), *z*(0), *w*(0)) = (−1, 0, 0, 0). Figure 1 shows the 2-D phase diagrams of the system, demonstrating the chaotic behavior: finiteness and instability.

Figure 2a displays the Lyapunov exponents (*LEs*) of the studied system based on the Wolf algorithm [37]. The results are *LE_1_* = 0.053671, *LE_2_* = −0.0049675, *LE_3_* = −0.099401 and *LE_4_* = −3.0757. *LE_1_* is positive, *LE_3_* and *LE_4_* are negative, *LE_2_* is nearly equal to zero and sum of the *LEs* is negative, that is, the whole phase volume of the studied system is exponentially shrinking. Figure 2b is the Poincaré map [38,39,40] of proposed system that contains random location of dots indicates the state of chaos. In brief, it is a dissipative chaotic system.

## 3. Dynamical Properties of the System

### 3.1. The Impacts of Parameters

In this section, the dynamic characteristics of the proposed system are discussed through varying the parameters *a*, *b* and *k* with the initial values as (*x*(0), *y*(0), *z*(0), *w*(0)) = (−1, 0, 0, 0). The numerical methods such as attractor phase diagrams, bifurcation diagram, Largest Lyapunov exponent spectrum and Spectral Entropy are used to observe the impacts of parameters.

In this paper, the Spectral Entropy algorithm, especially, is applied to analyze the impacts of parameters. It is briefly introduced as follows:(4)$$x(k)=\sum_{n\;=\;0}^{N-1}x(n)e^{-\frac{j2\pi nk}{N}}=\sum_{n\;=\;0}^{N-1}x(n)W_{N}^{nk},$$

In Equation (4), *x*(*n*) (*n* = 0, 1, 2…*N* − 1) is the pseudo-random sequence that has been removed the DC part and *x*(*k*) (*k* = 0, 1, 2…*N* − 1) is the discrete Fourier transform of *x*(*n*) (*n* = 0, 1, 2…*N* − 1). Based on Paserval algorithm, the power spectrum at some specific frequency can be obtained by the first half of the discrete sequence *x*(*k*) (*k* = 0, 1, 2…*N*/2 − 1).
(5)$$p(k)=\frac{1}{N}{\left|x(k)\right|}^{2}$$
and the total power spectrum can be described as:(6)$$p_{tot}(k)=\frac{1}{N}\sum_{k=0}^{\frac{N}{2}-1}{\left|x(k)\right|}^{2}$$

Combining Equations (5) and (6), the probability of the relative power spectrum *p_k_* can be expressed as *p*(*k*)/*p_tot_*(*k*). Based on the concept of Shannon entropy, the value of Spectral Entropy is equal to the sum of *p_k_* ln(1/*p_k_*) (*k* = 0, 1, 2…*N*/2−1). Because of the value of Spectral Entropy converging to ln(*N*/2), the normalized Spectral Entropy can be expressed as
(7)$$SE=\frac{-\sum_{k=0}^{\frac{N}{2}-1}p_{k}\ln p_{k}}{\ln(N/2)}$$

It can be seen from the above analysis that the bigger value of SE represents more complex oscillation for one chaotic system, otherwise, the smaller value of SE indicates more obvious oscillation for one chaotic system. Thus, we believe different values of SE should correspond to different states of the chaotic system.

Figure 3a presents the bifurcation diagram of the variable *z* versus the parameter *a* that has been achieved with setting the plane *y* = 0. Figure 3b shows the Largest Lyapunov exponents versus the parameter *a*. Figure 3c displays the Spectral Entropy versus the parameter *a*. When *a*∈[1.23, 1.35] and *a*∈[1.36, 1.43], the bifurcation diagram shows obvious chaotic states (see Figure 3a) and as expected the value of the Largest Lyapunov exponents are positive (see Figure 3b), correspondingly, the normalized values of SE are 0.4926 and 0.5032 (see Table 1) that are almost the same. When *a*∈[1.55, 1.6], the bifurcation diagram (see Figure 3a) exhibits quasiperiodic state and as expected the Largest Lyapunov exponent (see Figure 3b) is close to zero, correspondingly, the average value of SE is 0.2738 shown in Table 1. In general, there are two kinds of chaotic attractors displayed in Figure 4a, b and one kind of quasiperiodic attractor shown in Figure 4c in the range of *a*∈[1.2, 1.6]. Chaotic attractors with different types are provided with similar values of SE, while the value of SE with the quasiperiodic attractor is quite different from the one with the chaotic attractor.

When *b*∈[0.06, 0.16] and *b*∈[0.17, 0.4], both the bifurcation diagram (see Figure 5a) and the Largest Lyapunov exponents (see Figure 5b) show that the system is in the obvious chaotic state. Moreover, Figure 5b shows that there are two positive Lyapunov exponents in some ranges, indicating hyperchaotic behavior in the proposed system. Figure 5c shows the normalized average values of SE are 0.5151 and 0.5153 displayed in Table 1. Two types of chaotic attractors shown in Figure 6a,b further verify chaotic attractors with different types are almost with the same value of SE.

When *k*∈(0.10, 0.52), the bifurcation diagram (see Figure 7a) indicating the quasiperiodic state shows a better correlation with the Largest Lyapunov exponents (see Figure 7b) and in this region, the normalized average value of SE is 0.2056, as summarized in Table 1. When *k*∈(1.65, 2), both the bifurcation diagram (see Figure 7a) and the largest Lyapunov exponent (see Figure 7b) exhibit that the system is in the periodic state. At the same time, the normalized average value of SE is 0.1104, as shown in Table 1. The quasiperiodic attractor and the periodic attractor are separately presented in Figure 8a,b. Similarly, the quasiperiodic attractors with different types have almost the same value of SE. Moreover, from Table 1, it can be seen that the average value of SE with periodic attractor is the smallest compared with chaotic attractors and quasiperiodic attractors.

In this section, the simulation results of LLE and SE are obtained by the method of 4th-order Runge-Kutta’s with double precision using the time-step Δ*t* = 10^−1^ s and setting simulation time *T* = 500s. First, by respectively comparing the largest Lyapunov exponents (LLE) (see Figure 3b, Figure 5b and Figure 7b) and the Spectral Entropy (SE) (see Figure 3c, Figure 5c and Figure 7c), we can observe that the changing trend of the two calculation results are almost same. Moreover, when the hardware devices are selected as 32GB memory, Core i9-10900 CPU, the operating system is Windows 10, the calculation time of SE is about 1/160 of LLE, as listed in Table 2. The different parameters *a*, *b* and *k* are selected and different types of attractors from type I to type VII have been summarized in Table 1. Thus, based on the calculated results, the proposed system has the characteristics that the normalized average value of SE in the chaotic state is about 0.5, in the quasiperiodic state is about 0.2 and in the periodic state is about 0.1.

From Figure 9, Spectral Entropy distributions of the studied system are analyzed through varying two of the parameters *a*, *b*, *k* and different colors correspond to the magnitude of SE as shown in the color bar. Based on the above analysis shown in Table 1, it can be seen that the dark red color, SE value about 0.5, implies the state of chaos, the yellow color, SE value about 0.2, indicates quasi-periodic state and the light-yellow color, SE value about 0.1, indicates periodic state. The chaotic characteristic diagram based on SE algorithm is not only effective but also intuitive. Thus, in this paper, we propose to use this method to identify the states of the studied system.

### 3.2. Coexistence of Hidden Attractor

Coexistence of attractor is such a phenomenon that one chaotic system’s stable state changes from one to another with the same parameters but different initial values. In this section, the numerical methods of attractor phase diagrams, attractive basin and bifurcation diagram are used to analyse the particular phenomenon.

In Figure 10a, the parameters are fixed at *a* = 1.38, *b* = 1 and *k* = 1. The red phase diagram is obtained by setting the initial values as (*x*(0), *y*(0), *z*(0), *w*(0)) = (1, 0, 0, 0), while setting the initial values to (*x*(0), *y*(0), *z*(0), *w*(0)) = (−1, 0, 0, 0), the phase diagram changes to the blue one. To further investigate the characteristics, we changed this set of parameters to *a* = 1.55, *b* = 1 and *k* = 1. From Figure 10b, it can be seen that the red and blue phase diagrams are obtained, respectively, corresponding to the initial values of (*x*(0), *y*(0), *z*(0), *w*(0)) = (−1, 0, 0, 0) and (*x*(0), *y*(0), *z*(0), *w*(0)) = (1, 0, 0, 0). Moreover, From Figure 10a,b, we find that the two phase-diagrams have a certain symmetrical similarity.

In order to explore the phenomenon of the coexistence of attractors, the numerical methods of the attractive basin and bifurcation diagram are applied to the proposed system. The parameters *a* = 1.55, *b* = 1 and *k* = 1, the attractive basin in the initial plane of *x*(0)-*y*(0), has been drawn in Figure 11a. It includes two different color areas, indicating two types of attractors. The attractor in the red portrait, with the initial values (1, 0, 0, 0) shown in Figure 10b, is the typical attractor corresponding to the red color region of the attractive basin shown in Figure 11a, and the attractor in the blue portrait, with the initial values (−1, 0, 0, 0) shown in Figure 10b, is the typical attractor corresponding to the light blue colour region of attractive basin shown in Figure 11a. From Figure 11b, the red bifurcation portraits are under the initial condition of (1, 0, 0, 0) and the blue portraits are under the initial condition of (−1, 0, 0, 0). The two orbits present almost the same state but with different oscillation ranges, which also implies two types of attractors.

### 3.3. Controllability of Attractor

In order to investigate the controllability of the chaotic attractors of the proposed system, we set two parameters *p* and *q* in Equation (1) for the proposed system, as shown in Equation (8). As a result, when we vary the value of parameter *p*, the location of the chaotic attractor can be moved up and down as illustrated in Figure 12a. When we vary the value of parameter *q*, the shape of chaotic attractor can be zoomed out/in illustrated in Figure 12b. Thus, the chaotic attractor is controllable both in location and shape through adjusting the parameters.
(8)$$\left\{\begin{array}{l} \overset{•}{x}=y\\ \overset{•}{\dot{y}}=-x-yz\\ \overset{•}{z}=\left|x\right|+xy-1.35\\ \overset{•}{\omega }=x-q(w+p) \end{array}\right..$$

### 3.4. Transient Behaviour

Transient behavior is such a phenomenon that one chaotic system is unstable and will depart from one state to another at some point. In this section, this particular phenomenon is observed in the proposed system. We set the parameter *p* in Equation 2 as follows:(9)$$\left\{\begin{array}{l} \overset{•}{x}=ky\\ \overset{•}{y}=-x-yz\\ \overset{•}{z}=p\left|x\right|+xy-a\\ \overset{•}{\omega }=x-bw \end{array}\right.,$$

Let the parameters *a* = 0.1, *b* = 1, *k* = 1, *p* = 1, setting the initial values to (*x*(0), *y*(0), *z*(0), *w*(0)) = (−1, 0, 0, 0), selecting the calculation time as 5000 s. From Figure 13a, it can be seen that the time domain diagram is divided into two-time regions that shows the system has transient behavior. The Largest Lyapunov exponent (see Figure 13b) gradually decreases to zero with time *t*, showing the transient behavior from chaos to period. To future illustrate the transient behavior, the blue and red phase portrait shown in Figure 13c, respectively, correspond to the chaotic attractor and periodic attractor. The time-domain diagram in the time interval (1850 s, 2000 s) is displayed in Figure 13d.

Moreover, when the initial conditions are set as (*x*(0), *y*(0), *z*(0), *w*(0)) = (1, 1, 1, 1) and (*x*(0), *y*(0), *z*(0), *w*(0)) = (0, 0.8, −0.5, 0), the time domain diagram exhibits a different route from chaos to period as shown in Figure 14a,b, demonstrating the phenomenon of coexistence of attractors.

In summary, this section mainly illustrates transient behaviors. The numerical methods of the attractor phase diagram, time domain diagram and Largest Lyapunov exponent are given in detail and the results confirm the transition behavior and further demonstrates that the system has abundant dynamic characteristics.

## 4. Circuit Design

### 4.1. Improved Modular Circuit Design

The mathematical chaotic models can be implemented physically through electronic circuits, which now have been applied in various engineering applications such as RFID security systems [41], secure communications [27], robotics, and image encryption processing [42,43].

In this section, we need to convert the mathematical chaotic models into an electronic circuit. The variables *x*, *y*, *z* and *w* in Equation (2), respectively, corresponds to the voltages *Ux*, *Uy*, *Uz* and *Uw*, the parameter *a* to the voltage *Va*. Then, we obtain Equation (10).
(10)$$\left\{\begin{array}{l} \frac{dU_{x}}{dt}=U_{y}\\ \frac{dU_{y}}{dt}=-U_{x}-U_{y}U_{z}\\ \frac{dU_{z}}{dt}=\left|U_{x}\right|+U_{x}U_{y}-V_{a}\\ \frac{dU_{w}}{dt}=U_{x}-U_{w} \end{array}\right..$$

In order to better match the system, we set the time scale to be *τ*_0_ = 1/*RC* = 10^4^. The value of the resistor *R* and the capacitor *C* is, respectively, 10 kΩ and 10 nF. Then we define a new time variable *τ* instead of *t*, and *t* = *τ*_0_*τ*, then *dt* = *τ*_0_*dτ*. Thus, the new equation has been obtained as follows:(11)$$\left\{\begin{array}{l} \frac{dU_{x}}{d\tau }=\frac{1}{RC}U_{y}\\ \frac{dU_{y}}{d\tau }=\frac{1}{RC}(-U_{x}-U_{y}U_{z})\\ \frac{dU_{z}}{d\tau }=\frac{1}{RC}(\left|U_{x}\right|+U_{x}U_{y}-V_{a})\\ \frac{dU_{w}}{d\tau }=\frac{1}{RC}(U_{x}-U_{w}) \end{array}\right..$$

Based on the Kirchhoff’s laws, the circuit schematics are denoted in Figure 15. In the schematics we have used 10 pieces of TL083 operational amplifiers, 2 pieces of AD633 analog multipliers, 19 linear resistors, 4 capacitors, 2 diodes and the power supplies (±15*V_DC_*). Figure 15 shows the schematics of the chaotic system in which all the resistors *R_i_* = *R* = 10 kΩ (*i* = 1,2,3…19) and all the capacitors *C_i_* = *C* = 10 nF (*i* = 1,2,3,4).

### 4.2. Multisim Results

We used the software of Multisim 14.0 to build the circuit. The experimental results displayed in Figure 16a–f, with *V_a_* = 1.35 V, are almost the same as the numerical simulation results shown in Figure 1. Moreover, when setting *V_a_* = 0.1V, the circuit system has a transition from chaos (see Figure 17a) to period (see Figure 17b), which corresponds with the numerical simulation results shown in Figure 13c. Thus, we can reach the conclusion that the proposed 4-D system can be achieved physically.

## 5. Conclusions

In this work, a chaotic system has been constructed, which has hidden chaotic attractors without equilibrium. The dynamical properties of the chaotic system have been analyzed by several numerical methods. Particularly, the method of SE is much more effective in identifying the state of the designed system compared with LLE. Moreover, some attractive phenomena have been observed, including multiple attractors, coexistence attractor, controllability of attractor, transition behavior and the feasibility has been testified by an electronic circuit. It is obvious that the abundant dynamic characteristics is the source of randomness that may be used for chaos-based information security.

## 6. Discussion

In this work, we have recommended using Spectral Entropy algorithm to identify the state (chaos, quasi-period or period) of the studied system. However, the SE algorithm will perform improperly when the system shows multi-stability in some states (chaos, quasi-period or period). For instance, the phase diagrams in Figure 4a,b and Figure 6a,b show chaotic states but with different types that cannot be identified by SE algorithm, because they are provided with the similar SE values. Due to its high efficiency and limitations, SE algorithm can be used to preliminarily and quickly estimate the states of chaotic system. Moreover, the LLE algorithm also has its limitations, for example, in [44], since the studied system shows multi-stability, the inner normalization procedure of LLE algorithm leads to a periodic state where the system is chaotic. To avoid such errors, some other algorithms, such as phase diagrams, bifurcation diagram, time domain diagram and Spectral Entropy (SE) have been used to verify each other.

## Figures and Tables

**Figure 1 entropy-23-01341-f001:**
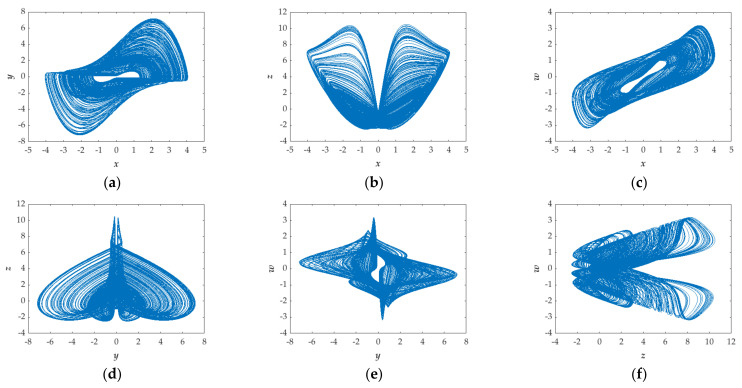
2-D phase portraits of the proposed system for *a* = 1.35, *b* = 1, *k* = 1 and the initial condition of (*x*(0), *y*(0), *z*(0), *w*(0)) = (−1, 0, 0, 0) in (**a**) *x*-*y* plane, (**b**) *x*-*z* plane, (**c**) *x*-*w* plane, (**d**) *y*-*z* plane, (**e**) *y*-*w* plane, and (**f**) *z*-*w* plane.

**Figure 2 entropy-23-01341-f002:**
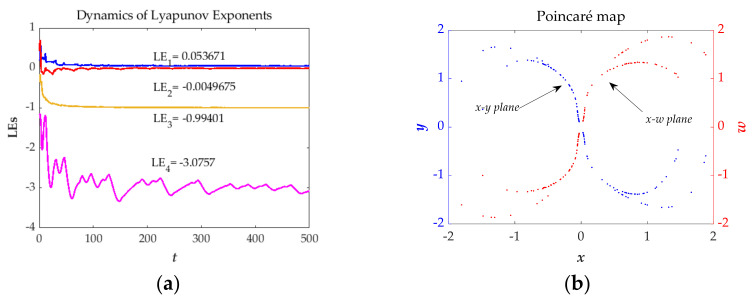
The dynamics of *LEs* (**a**) and Poincaré map (**b**) of the proposed system for *a* = 1.35, *b* = 1, *k* = 1 and the initial conditions (*x*(0), *y*(0), *z*(0), *w*(0)) = (−1, 0, 0, 0).

**Figure 3 entropy-23-01341-f003:**
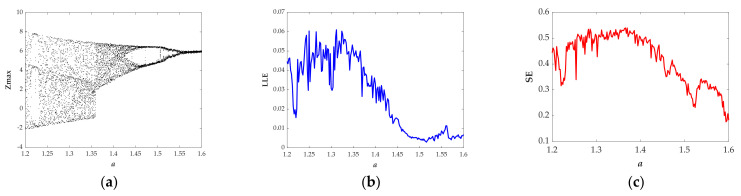
The evolution of the state descriptors of the proposed system when the parameter *a* changes in [1.2, 1.6] for the selected set of *b* = 1, *k* = 1 and the initial conditions (*x*(0), *y*(0), *z*(0), *w*(0)) = (−1, 0, 0, 0): (**a**) Bifurcation diagram; (**b**) Largest Lyapunov spectrum; (**c**) Spectral Entropy (Normalized).

**Figure 4 entropy-23-01341-f004:**
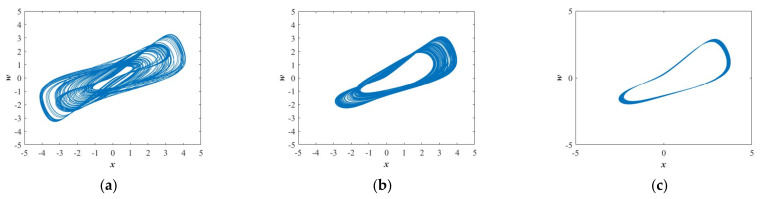
Projections of chaotic attractors in the *x*-*w* plane for the selected set of *b* = 1, *k* = 1, the initial conditions of (*x*(0), *y*(0), *z*(0), *w*(0)) = (−1, 0, 0, 0), and (**a**) *a* = 1.3; (**b**) *a* = 1.4; (**c**) *a* = 1.55.

**Figure 5 entropy-23-01341-f005:**
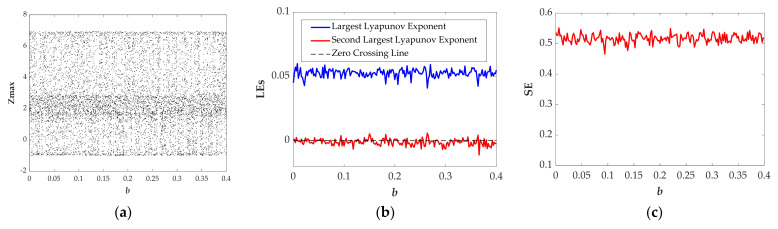
The evolution of the state descriptors of the proposed system when the parameter *b* changes in [0, 0.4] for the selected set of *a* = 1.35, *k* = 1 and the initial conditions (*x*(0), *y*(0), *z*(0), *w*(0)) = (−1, 0, 0, 0): (**a**) Bifurcation diagram; (**b**) Lyapunov Exponents; (**c**) Spectral Entropy(Normalized).

**Figure 6 entropy-23-01341-f006:**
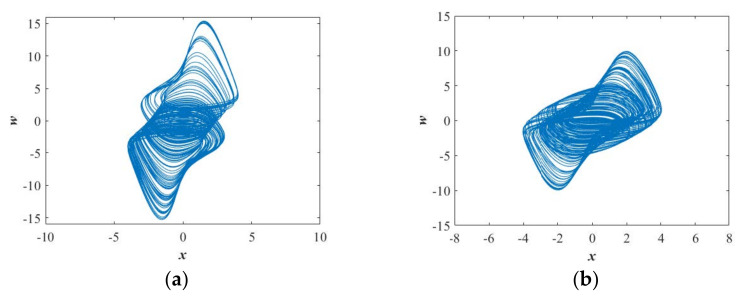
Projections of chaotic attractors in the *x*-*w* plane for the selected set of *a* = 1.35, *k* = 1, the initial conditions (*x*(0), *y*(0), *z*(0), *w*(0)) = (−1, 0, 0, 0), and (**a**) *b* = 0.1; (**b**) *b* = 0.3.

**Figure 7 entropy-23-01341-f007:**
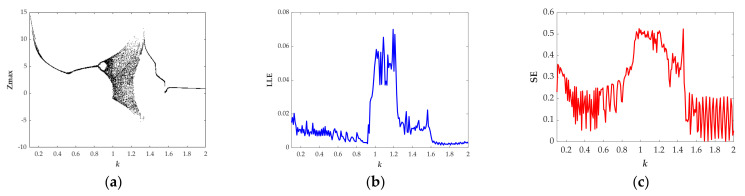
The evolution of state descriptors of the proposed system when the parameter *k* increases from 0.1 to 2.0 for the selected set of *a* = 1.35, *b* = 1 and the initial conditions (*x*(0), *y*(0), *z*(0), *w*(0)) = (−1, 0, 0, 0): (**a**) Bifurcation diagram; (**b**) Largest Lyapunov spectrum; (**c**) Normalized Spectral Entropy.

**Figure 8 entropy-23-01341-f008:**
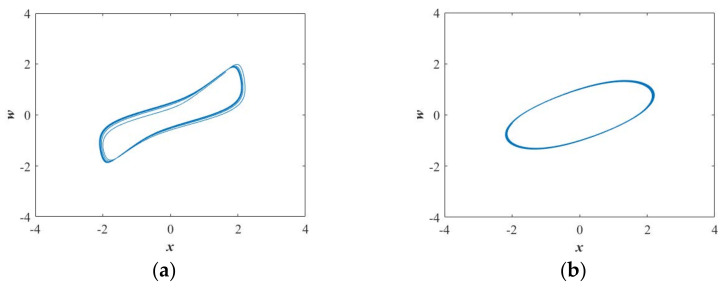
Projections of chaotic attractors in the *x*-*w* plane for the selected set of *a* = 1.35, *b* = 1, the initial conditions (*x*(0), *y*(0), *z*(0), *w*(0)) = (−1, 0, 0, 0), and (**a**) *k* = 0.3; (**b**) *k* = 1.8.

**Figure 9 entropy-23-01341-f009:**
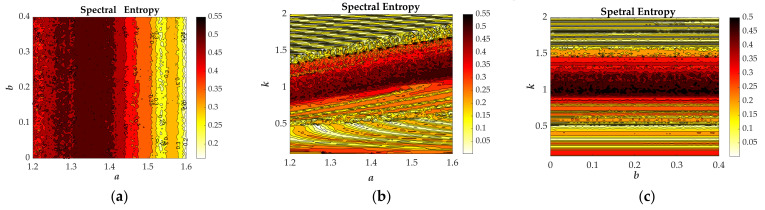
The chaotic characteristic diagram based on Spectral Entropy of the proposed system with the initial conditions (*x*(0), *y*(0), *z*(0), *w*(0)) = (−1, 0, 0, 0) (**a**) *a–b* plane with the *k* = 1, (**b**) *a–k* plane with the *b* = 1, (**c**) *b–k* plane with the *a* = 1.35.

**Figure 10 entropy-23-01341-f010:**
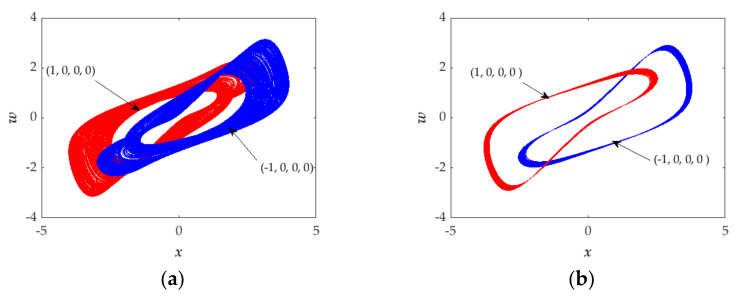
Projections of hidden attractors on *x*-*w* plane under the initial conditions of (1, 0, 0, 0) (blue) and (−1, 0, 0, 0) (red) with the different set of parameters: (**a**) *a* = 1.38, *b* = 1 and *k* = 1; (**b**) *a* = 1.55, *b* = 1 and *k* = 1.

**Figure 11 entropy-23-01341-f011:**
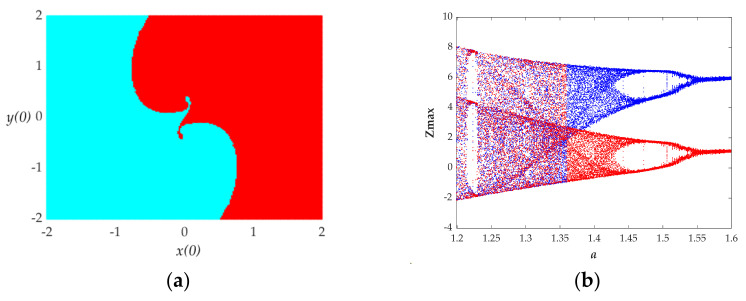
(**a**) Attractive basins with the parameter *a* = 1.55, *b* = 1, *k* = 1 and in the cross section of *z* (0) = 0 and *w* (0) = 0. (**b**) Bifurcation diagram under the initial condition of (1, 0, 0, 0) (red) and (−1, 0, 0, 0) (blue).

**Figure 12 entropy-23-01341-f012:**
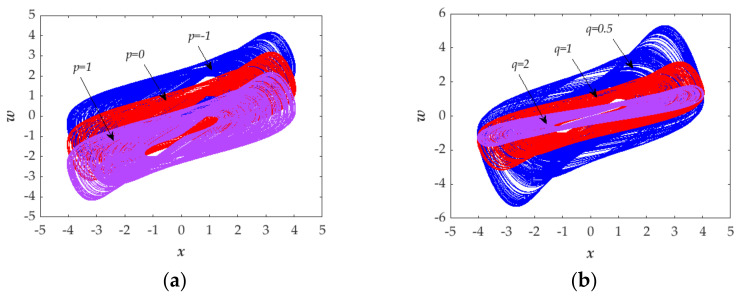
Controllability of chaotic attractor: (**a**) location controllable when adjusting the parameter *p* in *x*-*w* plane for *p* = 0 (red), *p* = 1 (purple), and *p* = −1 (blue); (**b**) shape controllable when adjusting the parameter *q* in *x*-*w* plane for *q* = 1 (red), *q* = 2 (purple), and *q* = 0.5 (blue).

**Figure 13 entropy-23-01341-f013:**
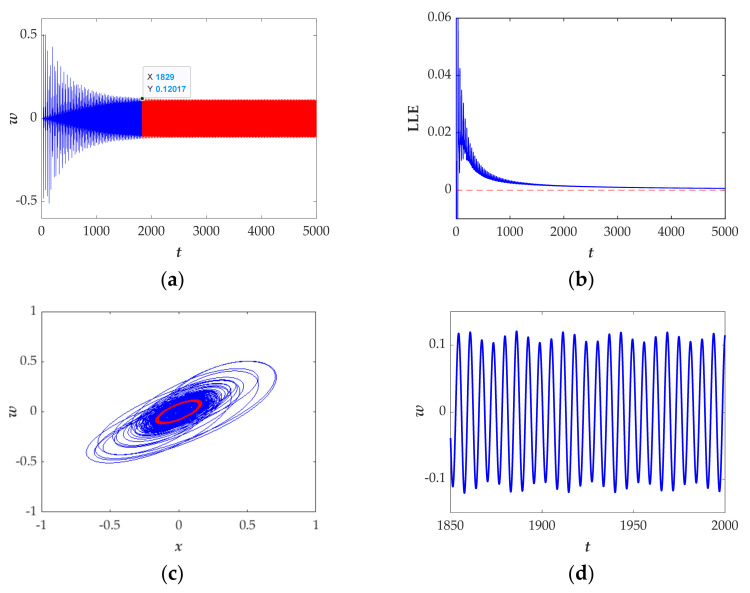
Transient behaviour from chaos to period with the selected set of *a* = 0.1, *b* = 1 and *k* = 1 and the initial conditions (*x*(0), *y*(0), *z*(0), *w*(0)) = (−1, 0, 0, 0): (**a**) time-domain diagram in the time interval [0 s, 5000 s]; (**b**) Largest Lyapunov exponent spectrum in the time interval [0 s, 5000 s]; (**c**) phase portrait from chaos (blue) to period (red); (**d**) time-domain diagram in the time interval [1850 s, 2000 s].

**Figure 14 entropy-23-01341-f014:**
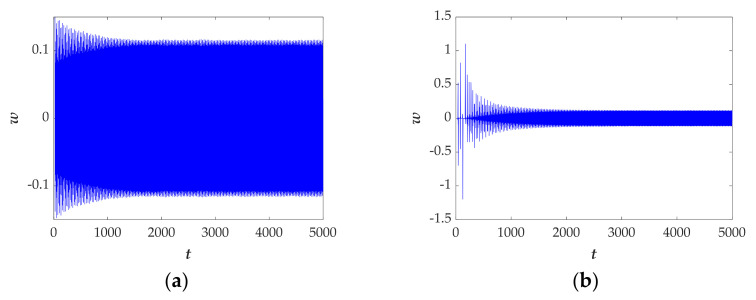
The time-domain diagram in the interval [0 s, 5000 s] with the selected set of *a* = 0.1, *b* = 1, *k* = 1, and the initial conditions of (*x*(0), *y*(0), *z*(0), *w*(0)) are set as (**a**) (1, 1, 1, 1), (**b**) (0, 0.8, −0.5, 0).

**Figure 15 entropy-23-01341-f015:**
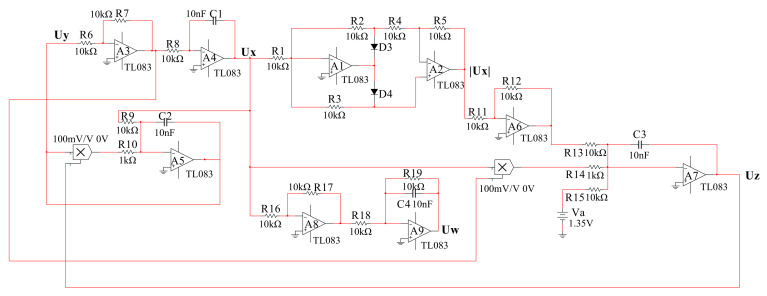
The schematics of the chaotic system.

**Figure 16 entropy-23-01341-f016:**
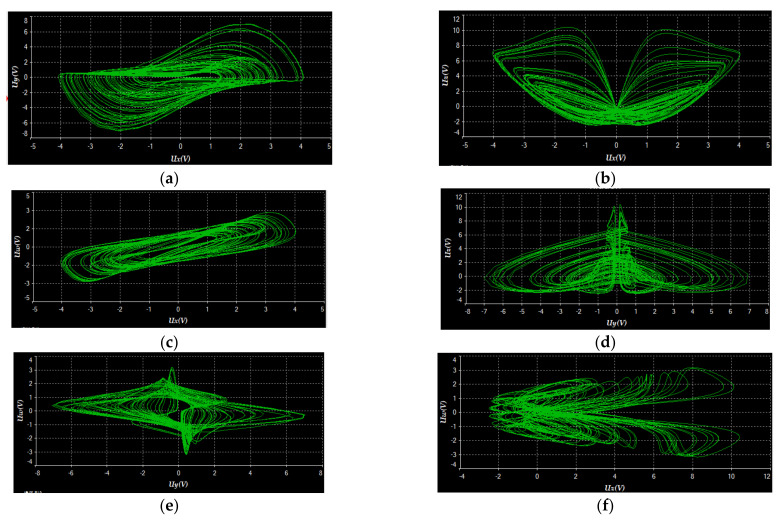
Experimental phase portraits with *V_a_* = 1.35 V displayed by using an oscilloscope: (**a**) *Ux*-*Uy* plane, (**b**) *Ux*-*Uz* plane, (**c**) *Ux*-*Uw* plane, (**d**) *Uy*-*Uz* plane, (**e**) *Uy*-*Uw* plane and (**f**) *Uz*-*Uw* plane.

**Figure 17 entropy-23-01341-f017:**
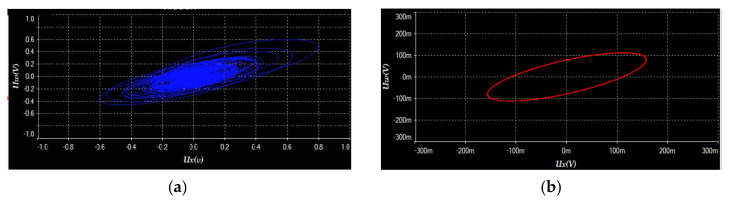
Experimental phase portraits with *V_a_* = 0.1 V displayed by using an oscilloscope: (**a**) chaotic attractor on *Ux*-*Uw* plane, (**b**) periodic attractor on *Ux*-*Uw* plane.

**Table 1 entropy-23-01341-t001:** States and SE values of the proposed system when the parameters *a*, *b* and *k* varying individually with the initial conditions (*x*(0), *y*(0), *z*(0), *w*(0)) = (−1, 0, 0, 0) and *a* = 1.35, *b* = 1, *k* = 1 when they are constant.

Range	State	Average Value of SE	Attractor Type	Corresponding Figure
*a*∈[1.23–1.35]	chaos	0.493	Ⅰ	Figure 4a
*a*∈[1.36–1.43]	chaos	0.502	Ⅱ	Figure 4b
*a*∈[1.55–1.6]	quasi-period	0.272	Ⅲ	Figure 4c
*b*∈[0.06–0.16]	chaos	0.513	Ⅳ	Figure 6a
*b*∈[0.17–0.40]	chaos	0.518	Ⅴ	Figure 6b
*k*∈[0.10–0.52]	quasi-period	0.208	Ⅵ	Figure 8a
*k*∈[1.65–2.00]	period	0.111	Ⅶ	Figure 8b

**Table 2 entropy-23-01341-t002:** Calculation time of LLE and SE for the three parameters *a*, *b*, and *k*.

Algorithm	*a* (s)	*b* (s)	*k* (s)
LLE	1384.948954	1383.054144	1353.828976
SE	8.153505	7.886318	9.132727

## Data Availability

Not applicable.

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
