# Peer review of "A Hidden Chaotic System with Multiple Attractors"

_entropy, 2021, doi:10.3390/e23101341_

Round 1

Reviewer 1 Report

In this paper, a chaotic system with hyper-phase space was suggested. Its Lyapunov exponent (LLE), bifurcation diagram, phase diagram, Poincaré map, time-domain waveform, attractive basin, and Spectral Entropy were presented. More, by changing altering the system parameter its chaotic attractor and transition behavior have also were studied. An electronic circuit modeled by a given chaotic system to verify the realizability.

Some comments:

1- There are some errors in references in some cases.

2-Eq 1, has strange notation. It must be corrected.

3- Authors may see the following paper to use this system in electrical circuits and add in the references:

Golmankhaneh, A.K., Arefi, R. and Baleanu, D., 2013. The proposed modified Liu system with fractional order. Advances in Mathematical Physics, 2013.

Reviewer 2 Report

  • There are major errors in typesetting references and equations on Word/LaTeX as well as redundant figure numbers, please revise.
  • The language and tenses also need revision.
  • Dozens of papers have followed [21] in the past few years studying the system or presenting variations on it. The authors should review these papers pointing out what exactly is novel in their manuscript.
  • Please provide more elaboration and evidence for “Compared with Viet-Thanh Pham and Christos Volos’s 3-D chaotic system, this 4-D system has more attractors, demonstrating better complexity.” Given that it is a 4-D system, are there any chances of hyperchaotic behavior?
  • In page 3, what do the authors mean by “thus it is a stable hidden chaotic system”? Dissipative?
  • More details about spectral entropy reference, equations and algorithm are required.
  • The formatting in Table 1 need some modifications in writing the range please use interval notation [ ] and mention the relation between attractor type and the figures as I barely noticed it.
  • SE is not “firstly” proposed in the paper as an alternative for LLE as implied in page 7, please check “Dynamics, synchronization and circuit implementation of a simple fractional-order chaotic system with hidden attractors,”, for example, refer to similar papers and modify the text accordingly:
  • The claim of obtaining a general law: “Thus, based on the calculated results, we can conclude that the normalized average value of SE in the chaotic state is about 0.5, in the quasiperiodic state is about 0.2 and in the periodic state is about 0.1.” cannot be only supported by simulations on a single system … These are just observations for this specific system and simulations not a general law …
  • More elaboration about the basin of attraction in Fig. 10 (a) and its color indications is needed.
  • The authors should suggest potential applications that require coexistence, controllability and transient behavior. It would be interesting to validate the transient behavior in the circuit simulation tool and identify the transient time as well.
  •  

Reviewer 3 Report

The peer-reviewed article describes a new chaotic system with hidden attractors. The results are interesting, but I cannot recommend the article for publication due to the following reasons:

1. Please do not use abbreviations in the abstract of the article.

2. References in the article are broken.

3. A comparison of the obtained system with the original model proposed by Viet-Thanh Pham and Christos Volos is required.

4. Recently, many models of systems with chaotic behavior have been proposed, including, for example, adaptive chaotic maps that exhibit the symmetry of the phase space. An overview of such unusual systems should be added to the introduction. In addition, it is worth mentioning a number of practical applications where such a system can be useful.

5. What numerical method was used to integrate the ODE system describing a chaotic system?

6. Could synchronization of digital and analog implementations of the proposed system be realized? Articles on this topic have also been recently published. How can hidden attractors affect the accuracy and speed of such synchronization?

7. In 2021, in the International Journal of Bifurcation and Chaos proposed a technique for detecting hidden oscillations in systems without equilibria through the estimation of the relative specific volume of the recurrence diagram was proposed. Please use this technique to detect attractors in the proposed system and show experimental results.

Thus, my decision is a major revision.

Reviewer 4 Report

The paper includes original results regarding the analysis of complex systems dynamics.

Numerical results are included. The results from a numerical point of view are good.

The real circuit implementation is emulated with a program like spice. Also in this case the results are suitable. Moreover I suppose that in the implementation with real case electronic circuits more appealing attractors could be found, The imperfections in electronics in some cases play a good role. I suggest to include the following reference in order to reinforce the previous item.

Imperfections in Integrated Devices Allow the Emergence of Unexpected Strange Attractors in Electronic Circuits

Authors

Maide Bucolo, Arturo Buscarino, Carlo Famoso, Luigi Fortuna, Salvina Gagliano

Publication date

2021/2/10

Journal

IEEE Access

Volume

9

Pages

29573-29583

Publisher

IEEE

Moreover  in real case systems It has been discovered strange and similar behaviour in systems appealing in mcrofluidic devices. It is clear in the following reference.

I suggest in order to make the paper more realistic to include the following paper.

ACNP Full Text

Nonlinear systems synchronization for modeling two-phase microfluidics flows

Authors

Fabiana Cairone, Princia Anandan, Maide Bucolo

Publication date

2018/4

Journal

Nonlinear Dynamics

Volume

92

Issue

1

Pages

75-84

Publisher

Springer Netherlands

Round 2

Reviewer 2 Report

No further comments.

Reviewer 3 Report

Dear Authors,

Thank you for taking my recommendations into account. I will not insist on changing the experimental part of the study, since the approaches used in the manuscript of the proposed chaotic system help to identify hidden oscillations in the model, as well as multiple attractors. However, I want to emphasize that some of the known approaches, including LLE, can give erroneous results when studying systems with multistability. For example, in the article that was mentioned in your last answer, it was shown that for the Sprott A system, the LLE metric defines periodic behavior when the system is chaotic. I find it essential to pay attention to this when choosing tools for studying new chaotic models. Therefore, I propose to discuss this issue in the final part of the article and mention this and other metrics that can be useful in the analysis of chaotic systems.
